# Online Learning under Adversarial Nonlinear Constraints

**Pavel Kolev**[1], **Georg Martius**[1,2], and **Michael Muehlebach**[1]

[1]Max Planck Institute for Intelligent Systems, Tübingen, Germany
[2]University of Tübingen, Tübingen, Germany
`{pavel.kolev, georg.martius, michael.muehlebach}@tuebingen.mpg.de`

## Abstract

In many applications, learning systems are required to process continuous non-stationary data streams. We study this problem in an online learning framework and propose an algorithm that can deal with adversarial time-varying and nonlinear constraints. As we show in our work, the algorithm called Constraint Violation Velocity Projection (CVV-Pro) achieves $\sqrt{T}$ regret and converges to the feasible set at a rate of $1/\sqrt{T}$, despite the fact that the feasible set is slowly time-varying and a priori unknown to the learner. CVV-Pro only relies on local sparse linear approximations of the feasible set and therefore avoids optimizing over the entire set at each iteration, which is in sharp contrast to projected gradients or Frank-Wolfe methods. We also empirically evaluate our algorithm on two-player games, where the players are subjected to a shared constraint.

## 1 Introduction

Today's machine learning systems are able to combine computation, data, and algorithms at unprecedented scales, which opens up new and exciting avenues in many domains, such as computer vision, computer graphics, speech and text recognition, and robotics [Jordan and Mitchell, 2015]. One of the leading principles that has enabled this progress is the focus on relatively simple pattern recognition and empirical risk minimization approaches, which mostly rely on offline gradient-based optimization and stipulate that the training, validation, and test data are independent and identically distributed.

Somewhat overlooked in these developments is the role of non-stationarity and constraints [Jordan, 2019]. Indeed, emerging machine learning problems involve decision-making in the real world, which typically includes interactions with physical, social, or biological systems. These systems are not only time varying and affected by past interactions, but their behavior is often characterized via fundamental constraints. Examples include cyber-physical systems where constraints are imposed by the laws of physics, multi-agent systems that are subjected to a shared resource constraint, or a reinforcement learning agent that is subjected to safety and reliability constraints. In particular, in their seminal work Auer et al. [2002] gave a reduction for the multi-arm bandit setting to the full information online optimization setting, by employing the multiplicative weights framework [Littlestone and Warmuth, 1994]. This classical reduction was recently extended by Sun et al. [2017] to the contextual bandit setting with sequential (time-varying) risk constraints.

This motivates our work, which is in line with a recent trend in the machine learning community towards online learning, adaptive decision-making, and online optimization. More precisely, we study an online problem with slowly time-varying constraints, governed by the following interaction protocol (see Assumption 1.2). In each time step $t$, the learner commits a decision $x_t$ and then in addition to a loss value $f_t(x_t)$ with its gradient $\nabla f_t(x_t)$ receives partial information about the current feasible set $\mathcal{C}_t := \{x \in \mathbb{R}^n \mid g_t(x) \geq 0\}$, where the constraint function $g_t(x)$ is defined as

37th Conference on Neural Information Processing Systems (NeurIPS 2023).

$[g_{t,1}(x), \ldots, g_{t,m}(x)]$. The quality of the learner's decision making is measured, for every $T \geq 1$, by comparing to the best decision in hindsight $x_T^\star \in \arg\min_{x \in \mathcal{C}_T} \sum_{t=1}^T f_t(x)$, that is,

$$\sum_{t=1}^T f_t(x_t) - \sum_{t=1}^T f_t(x_T^\star) \quad \text{subject to} \quad g_T(x_T) \geq -\frac{c}{\sqrt{T}}, \tag{1}$$

which will be shown to be bounded by $\mathcal{O}(\sqrt{T})$ for our algorithm. The functions $f_t$ and $g_t$ are restricted to $f_t \in \mathcal{F}$ and $g_t \in \mathcal{G}$ (as defined in Assumption 1.1) and $c > 0$ is an explicit constant.

It is important to note that our performance objective (1) is symmetric in the sense that the constraint $x \in \mathcal{C}_T$ applies to both the learner's decision $x_T$ and the benchmark $x_T^\star$. This contrasts prior work by Neely and Yu [2017], Yu et al. [2017], Sun et al. [2017], Chen et al. [2017], Cao and Liu [2019] and Liu et al. [2022], where a different notion of constraint violation $\sum_{t=1}^T g_t(x_t) \geq -c_0\sqrt{T}$ is used for the learner, while either a single benchmark $x_{1:T}^\star$ satisfies $g_t(x_{1:T}^\star) \geq 0$ for all $t \in \{1, \ldots, T\}$ or multiple benchmarks $\{x_t'\}_{t=1}^T$ satisfy $x_t' \in \arg\min_{x \in \mathcal{C}_t} f_t(x)$. Unlike (1), different requirements are imposed on the learner and the benchmark(s), which leads to asymmetric regret formulations: $\sum_{t=1}^T f_t(x_t) - \sum_{t=1}^T f_t(x_{1:T}^\star)$ and $\sum_{t=1}^T f_t(x_t) - \sum_{t=1}^T f_t(x_t')$, respectively. Furthermore, as our bound $g_T(x_T) \geq -c/\sqrt{T}$ applies for all $T \geq 1$, it implies the cumulative constraint violation bound in Neely and Yu [2017] up to a constant factor $\sum_{t=1}^T g_t(x_t) \geq -c \sum_{t=1}^T 1/\sqrt{t} \geq -2c\sqrt{T}$.

Even more intriguing is the fact that our algorithm is unaware of the feasible sets a-priori, and obtains, at each iteration, only a local sparse approximation of $\mathcal{C}_t$ based on the first-order information of the *violated* constraints. The indices of all violated constraints at $x_t$ will be captured by the index set $I(x_t) := \{i \in \{1, \ldots, m\} \mid g_{t,i}(x_t) \leq 0\}$, while $G(x_t) := [\nabla g_{t,i}(x_t)]_{i \in I(x_t)}$ denotes the matrix whose columns store the corresponding gradients. In order to guarantee a regret of $\mathcal{O}(\sqrt{T})$ in (1) we require the following assumptions.

**Assumption 1.1.** There exist $R, L_\mathcal{F}, L_\mathcal{G} > 0$: **1)** $\mathcal{F}$ is a class of convex functions, where every $f \in \mathcal{F}$ satisfies $||\nabla f(x)|| \leq L_\mathcal{F}, \forall x \in \mathcal{B}_{4R}$, with $|| \cdot ||$ the $\ell_2$ norm and $\mathcal{B}_R$ the hypersphere of radius $R$ centered at the origin; **2)** $\mathcal{G}$ is a class of concave $\beta_\mathcal{G}$-smooth functions, where every $g$ satisfies $||\nabla g(x)|| \leq L_\mathcal{G}, \forall x \in \mathcal{B}_{4R}$; **3)** The feasible set $\mathcal{C}_t$ is non-empty and contained in $\mathcal{B}_R$ for all $t$.

We note that these assumptions are standard in online optimization [Hazan, 2016, Ch. 3]. The learner's task is nontrivial even in the case where the feasible set is time invariant. If the feasible set is time varying, additional assumptions are required that restrict the amount that the feasible set is allowed to change. These two assumptions, see Part 2 i) and ii) below, are described by the following interaction protocol between the learner and the environment:

**Assumption 1.2.** (Interaction protocol) At each time step $t \in \{1, \ldots, T\}$:
**1)** the learner chooses $x_t$;
**2)** the environment chooses $f_t \in \mathcal{F}$ and $g_t \in \mathcal{G}$ such that i) $||g_t(x) - g_{t-1}(x)||_\infty = \mathcal{O}(1/t)$, uniformly for all $x \in \mathcal{B}_{4R}$, with $|| \cdot ||_\infty$ the $\ell_\infty$ norm, and ii) $\mathcal{C}_t$ is contained in $\mathcal{Q}_t := \cap_{\ell=0}^{t-1} \mathcal{S}_\ell$, where $\mathcal{S}_t := \{x \in \mathbb{R}^n \mid G(x_t)^\top (x - x_t) \geq 0\}$ is a cone centered at $x_t$ for $t \geq 1$ and $\mathcal{S}_0 = \mathbb{R}^n$ (the situation is illustrated in Figure 1, more details are presented in Appendix A);
**3)** the environment reveals to the learner partial information on cost $f_t(x_t), \nabla f_t(x_t)$ and all violated constraints $g_{t,i}(x_t), \nabla g_{t,i}(x_t)$ for $i \in I(x_t)$.

The requirements i) $||g_t - g_{t-1}||_\infty = \mathcal{O}(1/t)$; and ii) $\mathcal{C}_t \subset \mathcal{Q}_t$ restrict the feasible sets that the environment can choose. We note that despite the fact that $||g_t - g_{t-1}||_\infty = \mathcal{O}(1/t)$, $||g_1 - g_t||_\infty = \Theta(\ln(t))$, which means that the sequence of functions $g_t$ that defines $\mathcal{C}_t$ does *not* converge in general. As a result, $\mathcal{C}_t$ may evolve in such a way that the initial iterates $x_1, x_2, \ldots, x_{t_0}$ achieve a large cost compared to $\min_{x \in \mathcal{C}_T} \sum_{t=1}^T f_t(x)$, as these are constrained by the sets $\mathcal{C}_1, \mathcal{C}_2, \ldots, \mathcal{C}_{t_0}$, which may be far away from $\mathcal{C}_T$. The second requirement ii) $\mathcal{C}_t \subset \mathcal{Q}_t$ avoids this situation and is therefore key for obtaining an $\mathcal{O}(\sqrt{T})$ regret.

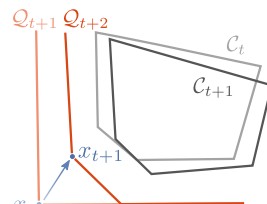

Figure 1: At each time step, the feasible set $\mathcal{C}_t$ contained in a polyhedral intersection $\mathcal{Q}_t$ changes slightly and is only partially revealed.

Our setup differs from traditional online convex optimization [Zinkevich, 2003] in the following two important ways:

i) The environment chooses not only the functions $f_t$ but also the nonlinear constraint functions $g_t$, ii) even if $g_t$ is time-invariant, i.e., $g_t = g$ for all $t$ the learner has only access to local information about the feasible set.

That is, the information about the feasible set is only revealed piece-by-piece and needs to be acquired by the agent through repeated queries of a constraint violation oracle.

We propose an online algorithm that despite the lack of information about the feasible set, achieves $\mathcal{O}(\sqrt{T})$ regret, and will derive explicit non-asymptotic bounds for the regret and the convergence to $\mathcal{C}_T$. We thus conclude that our algorithm matches the performance of traditional online projected gradients or Frank-Wolfe schemes, while requiring substantially less information about the feasible set and allowing it to be time-varying. Perhaps equally important is the fact that instead of performing projections onto the full feasible set at each iteration, our algorithm only optimizes over a local sparse linear approximation. If constraints are nonlinear, which includes norm-constraints or constraints on the eigenvalues of a matrix, optimizing over the full feasible set at each iteration can be computationally challenging.

## 1.1 Related Work

Online learning has its roots in online or recursive implementations of algorithms, where due to the piece-by-piece availability of data, algorithms are often analyzed in a non i.i.d. setting. A central algorithm is the multiplicative weights scheme [Freud and Schapire, 1997], where a decider repeatedly chooses between a finite or countable number of options with the aim of minimizing regret. This online learning model not only offers a unifying framework for many classical algorithms [Blum, 1998], but represents a starting point for online convex optimization Hazan [2016], and adversarial bandits [Lattimore and Szepesvári, 2020]. Our approach extends this line of work by allowing the environment to not only choose the objective functions $f_t$, but also the constraints $g_t$. Due to the fact that our learner only obtains local information about the feasible set, our work is somewhat related to Levy and Krause [2019], Lu et al. [2023], Garber and Kretzu [2022], Mhammedi [2022], where the aim is to reduce the computational effort of performing online projected gradient steps or Frank-Wolfe updates. More precisely, Levy and Krause [2019] propose an algorithm that directly approximates projections, while requiring multiple queries of the constraint functions and their gradients. A slightly different constraint violation oracle is assumed in Garber and Kretzu [2022], where the learner can query separating hyperplanes between a given infeasible point and the feasible set. Algorithmically, both Garber and Kretzu [2022] and Levy and Krause [2019] depart from online gradient descent, where the latter computes projections via an approximate Frank-Wolf-type scheme. An alternative is provided by Mhammedi [2022] and Lu et al. [2023], where optimizations over the entire feasible set are simplified by querying only a set membership oracle based on the Minkowski functional. While our approach also avoids projections or optimizations over the entire feasible set, we introduce a different constraint violation oracle that returns a local sparse linear approximation of the feasible set. We call the constraint violation oracle only once every iteration and do not require a two-step procedure that involves multiple oracle calls. In addition, we also allow for adversarial time-varying constraints.

In addition, there has been important recent work that developed online optimization algorithms with constraints. In contrast to the primal formulation of our algorithm, these works are based on primal-dual formulations, where the algorithm is required to satisfy constraints on average, so called long-term constraints. The research can be divided into two lines of work Mahdavi et al. [2012], Jenatton et al. [2016], Yu and Neely [2020] and Yuan and Lamperski [2018], Yi et al. [2021] that use a set of weaker and stricter definitions for constraint violations and investigate time-invariant constraints, which contrasts our formulation that includes time-varying constraints. A third line of work by Mannor et al. [2009], Chen et al. [2017], Neely and Yu [2017], Yu et al. [2017], Sun et al. [2017], Cao and Liu [2019], Liu et al. [2022] focuses on time-varying constraints, where, however, the following weaker notion of constraint violation is used: $\sum_{t=1}^{T} g_t(x_t) \geq -c\sqrt{T}$, where $t$ refers to time and $x_t$ to the learner's decision. This metric allows constraint violations for many iterations, as long as these are compensated by strictly feasible constraints (in the worst case even with a single feasible constraint with a large margin). In contrast, our algorithm satisfies $g_t(x_t) \geq -c/\sqrt{t}$ for all iterations $t \in \{1, \ldots, T\}$, where $c$ is an explicit constant independent of the dimension of the decision variable and the number of constraints. This means that we can explicitly bound the constraint violation at every iteration, whereas infeasible and strictly feasible iterates cannot compensate each other.

An important distinction to Neely and Yu [2017] is given by our performance metric (see also the discussion in Neely and Yu [2017] and Liu et al. [2022]). On the one hand, the work by Chen et al. [2017], Cao and Liu [2019], Liu et al. [2022] use $\sum_{t=1}^{T} f_t(x_t) - \sum_{t=1}^{T} f_t(x'_t)$ as a performance measure, where the iterates $x_t$ are required to satisfy $\sum_{t=1}^{T} g_t(x_t) \geq -c\sqrt{T}$ and the optimal solutions $x'_t$ satisfy $x'_t \in \arg\min_{x \in \mathcal{C}_t} f_t(x)$. On the other hand, the work by Neely and Yu [2017], Yu et al. [2017], Sun et al. [2017] use $\sum_{t=1}^{T} f_t(x_t) - \sum_{t=1}^{T} f_t(x^\star_{1:T})$ as a performance measure, where the iterates $x_t$ are required to satisfy $\sum_{t=1}^{T} g_t(x_t) \geq -c\sqrt{T}$ and the optimal solution $x^\star_{1:T}$ satisfies $g_t(x^\star_{1:T}) \geq 0$ *for all* $t \in \{1, \ldots, T\}$. This leads to a major asymmetry in the way regret is measured: while the iterates of the online algorithm only need to satisfy a cumulative measure of constraint violation, the benchmark $x^\star_{1:T}$, which represents the best fixed decision in hindsight, is required to satisfy *all* constraints $g_t(x^\star_{1:T}) \geq 0$ for $t = \{1, \ldots, T\}$. In contrast, the performance metric introduced in (1) is symmetric and imposes the same constraint $x \in \mathcal{C}_T$ (approximately) on the learner's decision $x_T$ and (exactly) on the benchmark $x^*_T$. These features make our algorithm a valuable addition to the algorithmic toolkit of online constrained optimization.

Castiglioni et al. [2022] studied the following asymmetric setting with adversarial environment, benchmark $x^\star_T$ belonging to $\arg\min_{x \in \mathcal{X}} \sum_{t=1}^{T} f_t(x)$ subject to $\frac{1}{T} \sum_{t=1}^{T} g_t(x) \geq 0$, online iterates $x_t$ satisfying $\frac{1}{T} \sum_{t=1}^{T} g_t(x_t) \geq -\mathcal{O}(1/\sqrt{T})$, and regret $\sum_{t=1}^{T} f_t(x_t) - \sum_{t=1}^{T} f_t(x^\star_T) \leq \mathcal{O}(\sqrt{T})$. Their benchmark and regret formulation can be obtained as a special case of our formulation with time-averaged constraints, that is, when our $g_T(x)$ is chosen as $\frac{1}{T} \sum_{t=1}^{T} g_t(x)$. In contrast, our iterate $x_T$ satisfies $\frac{1}{T} \sum_{t=1}^{T} g_t(x_T) \geq -\mathcal{O}(1/\sqrt{T})$, a constraint that is asymptotically the same as the one satisfied by the benchmark $x^\star_T$. We further note that they introduced a parameter $\rho = \sup_{x \in \mathcal{X}} \min_{t \in [T]} \min_{i \in [m]} g_{t,i}(x)$, which is required to be positive and *known* to the algorithm for achieving $\mathcal{O}(\sqrt{T})$ regret. Notably $\rho > 0$ implies that the intersection of all feasible sets is non-empty, which is a strong assumption (as is knowledge about the parameter $\rho$). In our formulation with *time-averaged* constraints, Assumption 1.2 reduces to the requirement that the feasible set $\mathcal{C}_t$ belongs to a polyhedral intersection $\mathcal{Q}_t$, which does not require a non-emtpy intersection of all $\mathcal{C}_t$ (has a geometrical interpretation and the assumption $\|g_t - g_{t-1}\|_\infty = \mathcal{O}(1/t)$ is automatically satisfied). Thus, there are situations, where the regret bound from Castiglioni et al. [2022] becomes vacuous (for $\rho = 0$), while our method still provably achieves $\mathcal{O}(\sqrt{T})$ regret. Additional differences are that Castiglioni et al. [2022] considers primal-dual methods and assumes that all constraints are revealed after every iteration, whereas our method is primal-only and has only partial information about all violated constraints. The latter point reduces computation and simplifies projections onto the velocity polyhedron, but requires a nontrivial inductive argument for establishing $\mathcal{O}(\sqrt{T})$ regret.

Other relevant related studies have investigated online learning problems with supply/budget constraints. In these settings, the decision maker must choose a sequence of actions that maximizes their expected reward while ensuring that a set of resource constraints are not violated. The process terminates either after a pre-specified time horizon has been reached or when the total consumption of some resource exceeds its budget. Badanidiyuru et al. [2018] introduced the bandits with knapsacks framework, which considers bandit feedback, stochastic objective and constraint functions. They proposed an optimal algorithm for this problem, which was later improved by Agrawal and Devanur [2014, 2019] and Immorlica et al. [2022]. Immorlica et al. [2022] introduced the adversarial bandits with knapsacks setting and showed that an appropriate benchmark for this setting is the best fixed distribution over arms. Since no-regret is no longer possible under this benchmark, they provide no-$\alpha$-regret guarantees for their algorithm.

An important special case of our online learning model arises when the environment is represented by an adversarial player that competes with the learner. This corresponds to a repeated generalized Nash game due to the constraint that couples the decisions of the learner and its adversary. If the adversary plays best response, the resulting equilibria are characterized by quasi-variational inequalities [Facchinei and Kanzow, 2007] and there has been important recent work, for example by Jordan et al. [2023], Kim [2021], Facchinei and Kanzow [2010] that proposes different gradient and penalty methods for solving these inequalities. Our approach adopts a different perspective, rooted in online learning, which allows us to derive non-asymptotic convergence results for a first-order gradient-based algorithm that can be implemented in a straightforward manner. Our approach is also inspired by the recent work of Muehlebach and Jordan [2022], who propose a similar algorithm for the offline setting.

## 1.2 Main Contributions

We give an online optimization scheme under *unknown* non-linear constraints that achieves an optimal $\mathcal{O}(\sqrt{T})$ regret and converges to the latest feasible set at a rate of $\mathcal{O}(1/\sqrt{T})$. There are two variants of our problem formulation: The first deals with situations where constraints are unknown but fixed, the second allows constraints to be chosen in a time-varying and adversarial manner.

Our algorithm, named Constraint Violation Velocity Projection (CVV-Pro), has the following features:

1. It assumes access to a new type of oracle, which on input $x_t$, returns partial information on all currently violated constraints. Namely, the value $g_{t,i}(x_t)$ and the gradient $\nabla g_{t,i}(x_t)$ for *all* $i \in I(x_t)$.

2. It projects an adversarially generated negative cost gradient $-\nabla f_t(x_t)$ onto a velocity polyhedron $V_\alpha(x_t) := \left\{ v \in \mathbb{R}^n \mid [\nabla g_i(x_t)]^\top v \geq -\alpha g_i(x_t), \ \forall i \in I(x_t) \right\}$. Due to the linear and local structure of $V_\alpha(x_t)$, the projection can be computed efficiently.

3. In contrast to standard online methods that project in each round a candidate decision onto the feasible set, our method trades off feasibility for efficiency. In particular, it produces a sequence of decisions that converges at a rate of $\mathcal{O}(1/\sqrt{T})$ to the latest feasible set.

4. Our method handles time-varying adversarial constraints $g_t$, provided a decreasing rate of change $||g_{t+1} - g_t||_\infty \leq \mathcal{O}(1/t)$ and that each feasible set $\mathcal{C}_t$ belongs to $\mathcal{Q}_t$ (see Assumption 1.2). As we show in Section 3.1, an important special case where the assumption of decreasing rate of change is satisfied is given by $g_t = \frac{1}{t} \sum_{j=1}^t \tilde{g}_j$, i.e., when $g_t$ represents an average of constraints $\tilde{g}_t$ over time.

## 1.3 Outline

Section 2 describes our algorithm and considers the situation where $g_t$ is time invariant. This sets the stage for our main results in Section 3 that provide regret guarantees for our new online convex optimization setting with non-stationary, nonlinear, and unknown constraints. An important and interesting application of our algorithm are generalized Nash equilibrium problems, as will be illustrated with a numerical experiment in Section 4. The experiment will also highlight that the numerical results agree with the theoretical predictions.

## 2 Online learning under unknown, time-invariant, and nonlinear constraints

### 2.1 Online Gradient Descent

Online gradient descent [Hazan, 2016, Ch. 3.1] is a classical and perhaps the simplest algorithm that achieves optimal $\mathcal{O}(\sqrt{T})$ regret for the setting of a compact, convex, time-invariant, and a priori known feasible set. It consists of the following two operations: i) $y_{t+1} = x_t - \eta_t \nabla f_t(x_t)$ takes a step from the previous point in the direction of the previous cost gradient; and ii) $x_{t+1} = \text{Proj}_{\mathcal{C}}(y_{t+1})$ projects $y_{t+1}$ back to the feasible set $\mathcal{C}$, as $y_{t+1}$ may be infeasible.

In this section, we generalize the online gradient descent algorithm to the setting where the feasible set is unknown a priori and has to be learned through repeated queries of a constraint violation oracle that only reveals local information.

### 2.2 Overview

In Section 2.3, we present the pseudo code of our algorithm. In Section 2.4, we give a structural result showing that Algorithm 1 under Assumption 1.1 and a bounded iterate assumption guarantees an optimal $\mathcal{O}(\sqrt{T})$ regret and converges to the feasible set at a rate of $\mathcal{O}(1/\sqrt{T})$. In Appendix E, we show that the bounded iterate assumption can be enforced algorithmically, by introducing an additional hypersphere constraint that attracts the sequence $\{x_t\}_{t \geq 1}$ to a fixed compact set.

### 2.3 Constraint Violation Velocity Projection (CVV-Pro)

We present below the pseudocode of Algorithm 1 for a fixed horizon length $T$, as it is standard in the literature [Hazan, 2016]. However, we note that our algorithm is oblivious to the horizon length $T$, i.e., it can run for any number of iterations without knowing $T$ a priori.

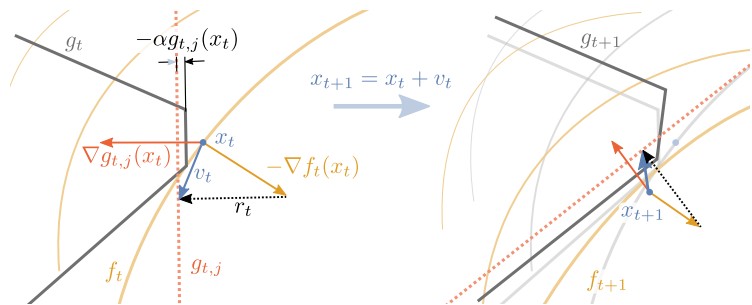

Figure 2: Illustration of the proposed (CVV-Pro) algorithm. **Left:** the constraint $g_{t,j}$ is violated by the current solution $x_t$. The cost gradient $-\nabla f_t(x_t)$ is projected onto the hyperplane (moved by $-\alpha g_{t,j}(x_t)$) with normal vector $\nabla g_{t,j}(x_t)$. This yields $r_t$ (see Section 2.5), and results in the velocity projection $v_t$ ($\eta = 1$ for clarity). **Right:** next iteration with updated $x$, where both $f$ and $g$ are changed. Then the procedure is applied recursively.

---

**Algorithm 1** Constraint Violation Velocity Projection (CVV-Pro)

---

1: **Requirements:** See Assumption 1.1
2: **Input:** $\alpha > 0$
3: **Initialization:** Step sizes $\left\{\eta_t = \frac{1}{\alpha\sqrt{t}}\right\}_{t \geq 1}$
4: **for** $t = 1$ **to** $T$ **do**
5:     **Play** $x_t$ **and observe**:
6:     cost information $f_t(x_t), \nabla f_t(x_t)$ and constraint information $\left\{\left(g_i(x_t), \nabla g_i(x_t)\right)\right\}_{i \in I(x_t)}$
7:     **Construct the velocity polyhedron as follows:**

$$V_\alpha(x_t) := \left\{v \in \mathbb{R}^n \mid [\nabla g_i(x_t)]^\top v \geq -\alpha g_i(x_t), \ \forall i \in I(x_t)\right\},$$

8:     **Solve the velocity projection problem:** $v_t = \arg\min_{v \in V_\alpha(x_t)} \frac{1}{2}\|v + \nabla f_t(x_t)\|^2$
9:     **Update:** $x_{t+1} = x_t + \eta_t v_t$
10: **end for**

---

Let $x \in \mathcal{C}$ be an arbitrary decision. We show in Claim 2.2 that $\alpha(x - x_t) \in V_\alpha(x_t)$. Hence, the velocity polyhedron $V_\alpha(x_t)$ is always non-empty and well defined.

## 2.4 Structural Result

Here, we show that Algorithm 1 under Assumption 1.1 and a bounded iterate assumption, guarantees an optimal $\mathcal{O}(\sqrt{T})$ regret and converges to the feasible set at a rate $\mathcal{O}(1/\sqrt{T})$. The bounded iterate assumption will be removed subsequently, which however, will require a more complex analysis.

**Theorem 2.1** (Structural). *Suppose Assumption 1.1 holds and in addition $x_t \in \mathcal{B}_R$ for all $t \in \{1, \ldots, T\}$. Then, on input $\alpha = L_\mathcal{F}/R$, Algorithm 1 with step sizes $\eta_t = \frac{1}{\alpha\sqrt{t}}$ guarantees the following for all $T \geq 1$:*

*(regret)*      $\sum_{t=1}^T f_t(x_t) - \min_{x \in \mathcal{C}} \sum_{t=1}^T f_t(x) \leq 18 L_\mathcal{F} R \sqrt{T}$;

*(feasibility)*   $g_i(x_t) \geq -8 \left[\frac{L_\mathcal{G}}{R} + 2\beta_\mathcal{G}\right] \frac{R^2}{\sqrt{t}}, \quad$ *for all $t \in \{1, \ldots, T\}$ and $i \in \{1, \ldots, m\}$.*

## 2.5 Proof Sketch of Theorem 2.1

Our analysis establishes, in two steps, an important geometric property that connects the convex costs and the concave constraints via the velocity polyhedron $V_\alpha(x_t)$. More precisely, we show that the inner product $-r_t^\top(x_T^\star - x_t) \leq 0$ for all $t \in \{1, \ldots, T\}$. This property will be crucial for deriving the regret and feasibility bounds.

In the first step, we leverage the constraints' concavity and show that the vector $\alpha(x_T^\star - x_t)$ belongs to the velocity polyhedron $V_\alpha(x_t)$.

**Claim 2.2.** *Suppose $g_i$ is concave for every $i \in \{1, \ldots, m\}$. Then $\alpha(x - x_t) \in V_\alpha(x_t)$ for all $x \in \mathcal{C}$. In addition, $x_t \notin \text{int}(\mathcal{C})$ implies $[\nabla g_i(x_t)]^\top[x - x_t] \geq 0$ for all $x \in \mathcal{C}$.*

*Proof.* Let $x \in \mathcal{C}$ be an arbitrary feasible decision, satisfying $g_i(x) \geq 0$ for all $i \in \{1, \ldots, m\}$. Since $g_i$ is concave, we have $g_i(x_t) + [\nabla g_i(x_t)]^\top [x - x_t] \geq g_i(x) \geq 0$ and thus $[\nabla g_i(x_t)]^\top [x - x_t] \geq -g_i(x_t)$. The second conclusion follows by $x_t \notin \mathrm{int}(\mathcal{C})$, which implies $g_i(x_t) \leq 0$. $\square$

In the second step, we show that $r_t^\top (x_t - x^\star) \leq 0$, where $r_t = v_t + \nabla f_t(x_t)$ is such that $-r_t$ belongs to the normal cone $N_{V_\alpha(x_t)}(v_t)$ of the velocity polyhedron $V_\alpha(x_t)$ evaluated at the projection $v_t$.

**Lemma 2.3** (Main). *Let $v_t$ be the projection of $-\nabla f_t(x_t)$ onto the polyhedron $V_\alpha(x_t)$ such that $v_t = r_t - \nabla f_t(x_t) \in V_\alpha(x_t)$, where $-r_t \in N_{V_\alpha(x_t)}(v_t)$. Then, $-r_t^\top (x - x_t) \leq 0$ for all $x \in \mathcal{C}$.*

*Proof.* By definition, the normal cone $N_{V_\alpha(x_t)}(v_t)$ is given by $\{u \in \mathbb{R}^n \mid u^\top (v - v_t) \leq 0, \ \forall v \in V_\alpha(x_t)\}$. Then, by construction $-r_t \in N_{V_\alpha(x_t)}(v_t)$ and thus it holds for every $v \in V_\alpha(x_t)$ that $-r_t^\top [v - v_t] \leq 0$. The proof proceeds by case distinction:

**Case 1.** Suppose $x_t$ is in the interior of $\mathcal{C}$. Then, $I(x_t) = \emptyset$, which implies $-\nabla f_t(x_t) \in V_\alpha(x_t) = \mathbb{R}^n$ and thus $r_t = 0$.

**Case 2.** Suppose $x_t$ is on the boundary or outside of $\mathcal{C}$, i.e., $I(x_t) \neq \emptyset$. By Claim 2.2, we have $[\nabla g_i(x_t)]^\top [x - x_t] \geq 0$ for all $x \in \mathcal{C}$. By construction, $v_t \in V_\alpha(x_t)$ and thus $v(x) = v_t + x - x_t \in V_\alpha(x_t)$. The statement follows by applying $v = v(x)$ to $-r_t^\top [v - v_t] \leq 0$. $\square$

**Regret.** To establish the first conclusion of Theorem 2.1 (regret), we combine the preceding geometric property with the analysis of online gradient descent. Since $f_t \in \mathcal{F}$ is convex, we upper bound the regret in terms of the gradient of $f_t$, namely $\sum_{t=1}^T f_t(x_t) - f_t(x_T^\star) \leq \sum_{t=1}^T [\nabla f_t(x_t)]^\top (x_t - x_T^\star)$ and then we show that the following inequality holds

$$
\begin{aligned}
[\nabla f_t(x_t)]^\top (x_t - x_T^\star) - \frac{\eta_t}{2} \|v_t\|^2 &= r_t^\top (x_t - x_T^\star) + \frac{\|x_t - x_T^\star\|^2 - \|x_{t+1} - x_T^\star\|^2}{2\eta_t} \\
&\leq \frac{\|x_t - x_T^\star\|^2 - \|x_{t+1} - x_T^\star\|^2}{2\eta_t}.
\end{aligned}
\tag{2}
$$

Moreover, in Appendix D (see Lemma D.2), we upper bound the velocity $\|v_t\| \leq \alpha \|x_T^\star - x_t\| + 2\|\nabla f_t(x_t)\|$. Combining Assumption 1.1 and $x_t \in \mathcal{B}_R$ yields a uniform bound $\|v_t\| \leq \mathcal{V}_\alpha$, where for $\alpha = L_\mathcal{F}/R$ we set $\mathcal{V}_\alpha := 4L_\mathcal{F}$. The desired regret follows by a telescoping argument and by convexity of the cost functions $f_t \in \mathcal{F}$.

**Feasibility.** For the second conclusion of Theorem 2.1 (convergence to the feasible set at a rate of $1/\sqrt{T}$), we develop a non-trivial inductive argument that proceeds in two steps. In Appendix D (see Claim D.6), we give a structural result that bounds the constraint functions from below. In particular, for every $i \in I(x_t)$ we have $g_i(x_{t+1}) \geq (1 - \alpha\eta_t)g_i(x_t) - \eta_t^2 \mathcal{V}_\alpha^2 \beta_\mathcal{G}$ and for every $i \notin I(x_t)$ it holds that $g_i(x_{t+1}) \geq -\eta_{t+1}\mathcal{V}_\alpha[2L_\mathcal{G} + \mathcal{V}_\alpha \beta_\mathcal{G}/\alpha]$.

Using an inductive argument, we establish in Appendix D (see Lemma D.5) the following lower bound: $g_i(x_t) \geq -c\eta_t$ where $c = 2\mathcal{V}_\alpha(L_\mathcal{G} + \mathcal{V}_\alpha \beta_\mathcal{G}/\alpha)$. Choosing $\alpha = L_\mathcal{F}/R$ implies that $\mathcal{V}_\alpha = 4L_\mathcal{F}$. Then, the desired convergence rate to the feasible set follows for the step size $\eta_t = \frac{1}{\alpha\sqrt{t}}$, since

$$
-c\eta_t = -\frac{2\mathcal{V}_\alpha}{\alpha\sqrt{t}}\left[L_\mathcal{G} + \frac{\beta_\mathcal{G}\mathcal{V}_\alpha}{\alpha}\right] = -8\left[\frac{L_\mathcal{G}}{R} + 4\beta_\mathcal{G}\right]\frac{R^2}{\sqrt{t}}.
$$

## 3 Online Learning under Adversarial Nonlinear Constraints

### 3.1 Problem Formulation

In this section, we consider an online optimization problem with adversarially generated time-varying constraints. More precisely, at each time step $t$, the learner receives partial information on the current cost $f_t$ and feasible set $\mathcal{C}_t$, and seeks to minimize (1). To make this problem well posed, we restrict the environment such that each feasible set $\mathcal{C}_t$ is contained in $\mathcal{Q}_t$ (see Section 1) and the rate of change between consecutive time-varying constraints *decreases* over time. We quantify a sufficient rate of decay with the following assumption.

**Assumption 3.1** (TVC Decay Rate). We assume that the adversarially generated sequence $\{g_t\}_{t\geq 1}$ of time-varying constraints is such that for every $x \in \mathcal{B}_{4R}$ and all $t \geq 1$, the following holds $\|g_{t+1}(x) - g_t(x)\|_\infty \leq \frac{98}{t+16}\left[\frac{L_\mathcal{G}}{R} + 3\beta_\mathcal{G}\right]R^2$.

We note that Assumption 3.1 essentially only requires $\|g_{t+1}(x) - g_t(x)\|_\infty \leq \mathcal{O}(1/t)$, as $R$ can be chosen large enough such that the bound is satisfied. Of course, $R$ will appear in our regret and feasibility bounds, but it will not affect the dependence on $t$ or $T$ (up to constant factors).

An important special case where Assumption 3.1 is satisfied, is summarized in the following Lemma. The proof is included in Appendix F (see Lemma F.7 and Lemma F.8).

**Lemma 3.2.** *Suppose the functions $\tilde{g}_{t,i}$ satisfy Assumption 1.1 and in addition there is a decision $x_{t,i} \in \mathcal{B}_R$ such that $\tilde{g}_{t,i}(x_{t,i}) = 0$ for every $t \geq 1$ and $i \in \{1, \ldots, m\}$. Then the time-averaged constraints $g_{t,i}(x) := \frac{1}{t}\sum_{\ell=1}^{t} \tilde{g}_{\ell,i}(x)$ satisfy Assumption 1.1 and Assumption 3.1.*

### 3.2 Velocity Projection with Attractive Hypersphere Constraint

We show in Appendix E that the second assumption in Theorem 2.1, namely, "$x_t \in \mathcal{B}_R$ for all $t \geq 1$" can be enforced algorithmically. We achieve this in two steps.

1) Algorithmically, we introduce an additional hypersphere constraint $g_{m+1}(x_t) = \frac{1}{2}[R^2 - \|x_t\|^2]$ that attracts the decision sequence $\{x_t\}_{t\geq 1}$ to a hypersphere $\mathcal{B}_R$ and guarantees that it always stays inside a hypersphere $\mathcal{B}_{4R}$ with a slightly larger radius.

More precisely, we augment the velocity polyhedron in Step 3 of Algorithm 1 as follows: $V'_\alpha(x_t) = V_\alpha(x_t)$ if $\|x\| \leq R$, otherwise

$$V'_\alpha(x_t) = \{v \in V_\alpha(x_t) \mid [\nabla g_{m+1}(x_t)]^\top v \geq -\alpha g_{m+1}(x_t)\}.$$

2) Analytically, we give a refined inductive argument in Appendix F (see Lemma E.5), showing that $g_{m+1}(x_t) \geq -27R^2/\sqrt{t + 15}$, $\|x_t\| \leq 4R$ and $\|v_t\| \leq 7L_\mathcal{F}$, for all $t \geq 1$.

### 3.3 Main Contribution

Our main contribution is to show that Algorithm 1 with the augmented velocity polyhedron $V'_\alpha(x_t)$, achieves optimal $\mathcal{O}(\sqrt{T})$ regret and satisfies $g_T(x_T) \geq -\mathcal{O}(1/\sqrt{T})$ convergence feasibility rate. Due to space limitations, we defer the proof to Appendix F.

**Theorem 3.3** (Time-Varying Constraints). *Suppose the functions $\{f_t, g_t\}_{t\geq 1}$ satisfy Assumptions 1.1, 1.2 and 3.1. Then, on input $R, L_\mathcal{F} > 0$ and $x_1 \in \mathcal{B}_R$, Algorithm 1 applied with $\alpha = L_\mathcal{F}/R$, augmented velocity polyhedron $V'_\alpha(\cdot)$ and step sizes $\eta_t = \frac{1}{\alpha\sqrt{t+15}}$ guarantees the following for all $T \geq 1$:*

*(regret)* $\quad \sum_{t=1}^{T} f_t(x_t) - \min_{x\in\mathcal{C}_T}\sum_{t=1}^{T} f_t(x) \leq 246 L_\mathcal{F} R\sqrt{T};$

*(feasibility)* $\quad g_{t,i}(x_t) \geq -265\left[\frac{L_\mathcal{G}}{R} + 4\beta_\mathcal{G}\right]\frac{R^2}{\sqrt{t+15}}, \quad$ *for all $t \in \{1, \ldots, T\}$ and $i \in \{1, \ldots, m\}$;*

*(attraction)* $\quad g_{m+1}(x_t) \geq -27\frac{R^2}{\sqrt{t+15}}, \quad$ *for all $t \in \{1, \ldots, T\}$.*

Our regret analysis in Theorem 3.3 builds upon the following key structural result that generalizes Lemma 2.3 to time-varying constraints. In particular, in Appendix F (see Lemma F.3), we show that given the feasible set $\mathcal{C}_T \subset \mathcal{Q}_T$, it holds for every $x \in \mathcal{C}_T$ that $-r_t^\top(x - x_t) \leq 0$ for all $t \in \{1, \ldots, T\}$. As a result, a similar argument as in (2) shows that the regret is bounded by $\mathcal{O}(\sqrt{T})$.

Moreover, we note that the linear and quadratic dependence on $R$ in Theorem 3.3 is consistent in length units. Let the radius $R$ be of length units $\ell$, then the Lipschitz constant $L_\mathcal{F}$, which can be viewed as the supremum over the $\ell_2$ norm of the gradient is of $1/\ell$ units, and the $\beta_\mathcal{G}$ smoothness constant (associated with Hessian) is of $1/\ell^2$ units. This means that the regret bound in Theorem 3.3 has the same units as $f_t$, while the feasibility bound has the same units as $g_t$.

## 4 Simulation examples

Two-player games with shared resources are an excellent example for demonstrating the effectiveness and importance of our online learning framework. We apply our algorithm and show numerical experiments that support our theoretical findings.

We choose random instances of a two player game with linear utility and constraints. In particular, we consider the following optimization problem

$$\min_{x\in\triangle_n} \max_{y\in\triangle_n} x^\top A y \quad \text{subject to} \quad C_x x + C_y y \leq 1, \tag{3}$$

where $\triangle_n = \{x \in \mathbb{R}^n \mid \sum_{i=1}^n x_i = 1, x \geq 0\}$ is the probability simplex. Each component of the utility matrix $A \in \mathbb{R}^{n\times n}$ is sampled from the normal distribution and the constraint matrices $C_x, C_y \in [0,1]^{m\times n}$ have each of their components sampled uniformly at random from $[0,1]$.

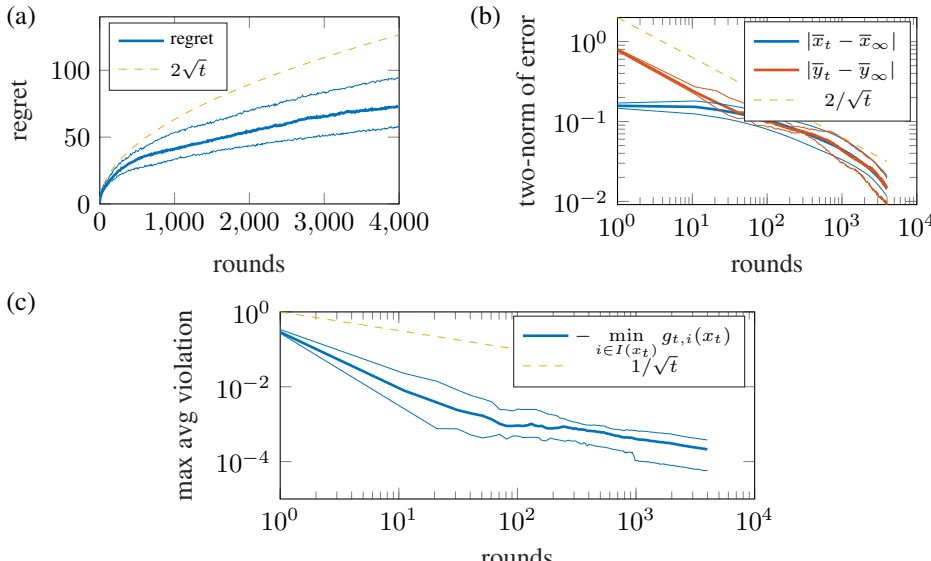

Figure 3: (**Setup**) The CVV-Pro algorithm is executed on five random instances of the two-player game with shared resources (Section 4.1). The thick line is the mean and the thin lines indicate the minimum and the maximum over the five runs. (a) The regret follows the predicted $\mathcal{O}(\sqrt{T})$ slope. (b) The CVV-Pro algorithm achieves a convergence rate of $\mathcal{O}(1/\sqrt{t})$ for the averaged decisions $\overline{x}_t := \frac{1}{t}\sum_{\ell=1}^{t} x_\ell$ towards $\overline{x}_\infty$. In our experiment, we set $\overline{x}_\infty := \overline{x}_{10000}$. Similar behavior is reported for the averaged decisions $\overline{y}_t$ of the adversary. (c) The maximal constraint violation expressed by $-\min_{i\in I(x_t)} g_{t,i}(x_t)$ converges at a rate of $\mathcal{O}(1/\sqrt{t})$, as predicted by our theoretical results. (**Compute**) For the implementation of CVV-Pro we have used the MATLAB R2019a numerical computing software. The computation of the experiment takes about 4 hours on a machine with CPU: Intel(R) i7-6800K 3.40 GHz with 6 cores, GPU: NVIDIA GeForce GTX 1080, and RAM: 32 GB.

## 4.1 Online Formulation

The problem in (3) can be modeled with our online learning framework (1) by choosing costs $f_t(x) := x^\top A y_t$ and time-averaged resource constraints $g_T(x) := \frac{1}{T}\sum_{t=1}^{T} \widetilde{g}_t(x)$, where the function $\widetilde{g}_t(x) := 1 - C_x x - C_y y_t$. Thus, the constraint in (3) is included as an average over the past iterations of $y_t$. The strategy for choosing $y_t$ will be described below and, as we will see, the average of $y_t$ over the past iterations converges. This ensures that the feasible set $\mathcal{C}_t$ (defined in (1)) is slowly time-varying, while the averages of $x_t$ and $y_t$ over past iterates converge to equilibria in (3). Further, by a refined version of Lemma 3.2 (see Lemma F.6 in Appendix F), the time-averaged constraints $g_T(x)$ satisfy Assumption 3.1.

In each iteration, Algorithm 1 seeks to minimize the online problem and commits to a decision $x_t$. The adversary computes the best response $\hat{y}_t$ with respect to the decision $x_t$ by solving $\arg\max_{y\in\triangle_n} x_t^\top A y$. To make the dynamics more interesting, the adversary then commits with probability 0.8 to $\hat{y}_t$ and with probability 0.2 to a random decision $\xi_t$, i.e., $y_t = 0.8\hat{y}_t + 0.2\xi_t$ where the random variable $\xi_t$ is sampled uniformly at random from the probability simplex $\triangle_n$.

As both players optimize over the probability simplex ($x, y \in \triangle_n$), the sequence of decisions $\{x_t\}_{t\geq 1}$ is automatically bounded. Thus, we can apply Theorem 3.3 with the original velocity polyhedron, as discussed in Appendix E. We implemented our algorithm with $\eta_t = 1/(\alpha\sqrt{t})$ and $\alpha = 100$.

## 4.2 Experimental Results

We report results from numerical simulations with decision dimension $n = 100$, $m = 10$ shared resource constraints, $T = 4000$ iterations, and five independently sampled instances of the two-player game. The learner's regret, depicted in Figure 3a, shows a clear correspondence with the theoretical prediction of $\mathcal{O}(\sqrt{T})$. Figure 3b presents the maximal constraint violation $-\min_{i \in I(x_T)} \frac{1}{T} \sum_{t=1}^{T} \widetilde{g}_{t,i}(x_T)$, which follows the predicted $\mathcal{O}(1/\sqrt{T})$ convergence rate. We also conclude from Figure 3c that the learner's averaged decisions $\overline{x}_T = \frac{1}{T} \sum_{t=1}^{T} x_t$ converge at a rate of $\mathcal{O}(1/\sqrt{T})$. Similarly, the averaged decisions $\overline{y}_T$ of the adversary also converge at a rate of $\mathcal{O}(1/\sqrt{T})$. We note that there is little variability in the results despite the different realizations of the matrices $A, C_x, C_y$.

**Contrasting CVV-Pro and Online Gradient Descent**    In Appendix C, we show that our (CVV-Pro) algorithm outperforms the standard Online Gradient Descent algorithm in the two-player game from above. In particular, our algorithm achieves a lower regret and a runtime improvement of about $60\%$. Further, the percentage of violated constraints decreases rapidly and plateaus at $20\%$.

The amount of improvement in execution time is likely to be greater for higher-dimensional problems, where fewer constraints tend to be active at each iteration. Moreover, when the constraints are nonlinear, which includes $\ell_p$ norm or spectral constraints, optimizing over the full feasible set can be computationally challenging. In contrast, the velocity projection step in CCV-Pro is always a convex quadratic program with linear constraints, regardless of the underlying feasible set.

## 5 Broader Impact

It is important to emphasize that our work is theoretical, and the main contribution is to design and analyze a novel algorithm that combines techniques from the seemingly distant fields of online convex optimization (online gradient descent) and non-smooth mechanics (velocity space, see Muehlebach and Jordan [2022]). Nevertheless, the list of potential applications includes, but is not limited to: adversarial contextual bandits with sequential risk constraints Sun et al. [2017], network resource allocation Chen et al. [2017], logistic regression Cao and Liu [2019], Liu et al. [2022], ridge regression and job scheduling Liu et al. [2022], two-player games with resource constraints (Section 4), system identification and optimal control (Appendix B).

## 6 Conclusion

We propose an online algorithm that, despite the lack of information about the feasible set, achieves $\mathcal{O}(\sqrt{T})$ regret. We further ensure convergence of violated constraint $-\min\{g_T(x_T), 0\}$ at a rate of $\mathcal{O}(1/\sqrt{T})$ and derive explicit constants for all our bounds that hold for all $T \geq 1$. We thus conclude that our algorithm matches the performance of traditional online projected gradients or Frank-Wolfe schemes, while requiring substantially less information about the feasible set and allowing the feasible set to be time-varying. Perhaps equally important is the fact that instead of performing projections onto the full feasible set at each iteration, our algorithm only optimizes over a local sparse linear approximation. We show the applicability of our algorithm in numeric simulations of random two-player games with shared resources.

## Acknowledgements

We acknowledge the support from the German Federal Ministry of Education and Research (BMBF) through the Tübingen AI Center (FKZ: 01IS18039B). Georg Martius is a member of the Machine Learning Cluster of Excellence, EXC number 2064/1 – Project number 390727645. Pavel Kolev was supported by the Cyber Valley Research Fund and the Volkswagen Stiftung (No 98 571). Michael Muehlebach thanks the German Research Foundation and the Branco Weiss Fellowship, administered by ETH Zurich, for the support. We thank anonymous reviewers for comments, which helped improve the presentation of the paper.

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
