# OpenReview forum: "Online Learning under Adversarial Nonlinear Constraints"
_NeurIPS.cc/2023/Conference — NeurIPS 2023 poster_

### Official Review · Reviewer_pGFm · 2023-06-24

**Soundness:** 2 fair
**Presentation:** 1 poor
**Contribution:** 2 fair
**Rating:** 5
**Confidence:** 2

**Summary:**

This paper studies the online convex optimization subject to certain constrains and assumes that such constrains are slowly changing. The main contribution is a $O(\sqrt{T})$ regret bound under such constrains.

**Strengths:**

This paper appears to prove some interesting technical results. However, due to its unclear presentation, I find it challenging to fully appreciate the significance of these findings.

**Weaknesses:**

The presentation of this paper is very confusing, making it difficult to follow even in the introduction. Let me elaborate on the following points:

1. The purpose of introducing the constraints $\mathcal{C}_t$ for $t < T$ is unclear. From your regret formulation in (1), it appears to only depend on $\mathcal{C}_T$. Are you assuming that $x_t$ must be selected from $\mathcal{C}_t$? This doesn't seem to align with Assumption 2.
2. There is no clear motivation behind the definition of (1). What is the reason for setting the constraints $g(x_T)\ge c/\sqrt{T}$?
3. The set $I(x_t)$ is not well-defined. Where does $m$ originate from?
4. What is the rationale for assuming $||g_t-g_{t+1}||\le O(1/t)$? Why not $1/t^2$, $1/\log t$, or $1/e^t$? How would the results change under these different bounds?
5. What novel concept does Algorithm 1 introduce? Is it the construction of the velocity polytope?
6. Can you highlight the main technical originality in your proofs? They appear to be simple applications of the analysis of online gradient descent.
7. Could you provide some real-world examples where your results can be applied?

Considering the issues mentioned above, my recommendation leans towards a "borderline reject." The paper lacks clarity in presentation, and several assumptions and definitions are not adequately justified or explained.

**Questions:**

Unfortually, I do not have any interesting technical questions for the authors at this point.

**Limitations:**

No issue with negative societal impact.

---

> ### Author Rebuttal · Authors · 2023-08-09
>
> Thank you for your valuable suggestions and important remarks.
> We answer the raised questions below.
>
> **Weaknesses**
>
> 1. "The purpose of introducing the constraints $\mathcal{C}_t$ for $t<T$ is unclear. ... Are you assuming that $x_t$ must be selected from $\mathcal{C}_t$? ..."
>
> CVV-Pro optimizes (1) for every $T\geq1$, since $T$ is not a priori known to the learner.
> Moreover, the learner cannot ensure that $x_t\in\mathcal{C}_t$, since the environment reveals a feasible set $\mathcal{C}_t$ after $x_t$ is committed.
>
> 2. "There is no clear motivation behind the definition of (1). What is the reason for setting the constraints $g(x_T)\geq c/\sqrt{T}$"
>
> Our formulation in (1) is symmetric and imposes the same constraints on the learner as well as the optimal decision in hindsight.
> Since the optimal solution $x^{\star}\in\mathcal{C}\_T$ satisfies $g_T(x^{\star})\geq0$, we seek to compute a decision $x_T$ that is close to the feasible set $\mathcal{C}\_T$, quantitatively $g(x_T)\geq - c/\sqrt{T}$, while maintaining small regret.
> Furthermore, as our constraint violation bound $g_T(x_T)\geq - \frac{c}{\sqrt{T}}$ applies for every $T\geq1$, it implies the cumulative constraint violation bound in Liu et al. [2022]: $\sum_{t=1}^{T}g_{t}(x_{t})\geq-2c\sqrt{T}$.
> See our answer to question 1 and weakness 1 to (7SGB) for more details.
>
> 3. "The set $I(x_t)$ is not well-defined. Where does $m$ originate from?"
>
> Based on your comments and those of the other reviewers, we will improve the introduction accordingly.
>
> 4. "What is the rationale for assuming $\lVert g_t - g_{t+1} \rVert\leq O(1/t)$?
> Why not $1/t^2$, $1/\log(t)$, or $1/e^t$?
> How would the results change under these different bounds?"
>
> The short answer is: it balances a trade-off between the rate of convergence to the feasible set and the power of the environment to select feasible sets far away from the initial feasible set.
> It is important to note that the series $\sum_{i=1}^{n}1/i\approx\ln(n)$ diverges, whereas for any $\epsilon>0$ the series $\sum_{i=1}^{n}1/i^{1+\epsilon}$ converges to a constant.
> In particular, $\lVert g_t - g_{t+1} \rVert\leq O(1/t)$ allows the environment to select a constraint function $g_T$ whose corresponding feasible set is far away from the initial, in the sense that $\lVert g_1 - g_{T} \rVert = \Theta(\ln T)$.
> On the other hand, relaxing the upper bound to say $\lVert g_t - g_{t+1} \rVert\leq O(1/\sqrt{t})$ results in a slower than $1/\sqrt{T}$ convergence rate to the feasible set.
>
> 5. "What novel concept does Algorithm 1 introduce? Is it the construction of the velocity polytope?"
>
> The short answer is yes: the velocity polytope is one of the novelties. However there is more to it:
>
> a) We combine techniques from online convex optimization and non-smooth mechanics.
> More specifically, we substitute the computationally expensive projection problem in the online gradient descent algorithm with a sparse linear velocity projection problem that admits efficient computation;
>
> b) Our algorithm requires in each iteration only partial information about all violated constraints.
> In contrast, prior works need full information for all constraints;
>
> c) CVV-Pro handles time varying constraints, under the assumption that the constraints are slowly time-varying and the corresponding feasible sets belong to certain polyhedral intersections (see the Assumption part in our global response).
>
> 6. "Can you highlight the main technical originality in your proofs? They appear to be simple applications of the analysis of online gradient descent."
>
> We find the elegance of the proposed geometric arguments particularly appealing.
> A major technical originality is to prove that the velocity (the algorithm's increments) is bounded, which requires the addition of an (ingenious) ``auxiliary" constraint, and non-trivial inductive arguments.
>
> More specifically, the main technical contributions are listed below:
>
> **Time-invariant constraints**
>
> We show that the velocity projection problem is well defined, as
> the velocity polytope is always non-empty.
> Moreover, we give an important geometric property showing that the inner product $-r_{t}^T(x^{\star}-x_t)\leq0$ for all $t$.
> Assuming the velocity is bounded, we can apply the standard OGD analysis to obtain a $\sqrt{T}$ regret.
> Then, we give a non-trivial inductive argument that yields convergence to the feasible set at a rate of $1/\sqrt{T}$.
> To enforce a uniform bound on the velocity, we augment the velocity polytope with an attraction constraint, ensuring that the learner's decision sequence always remains within a hypersphere of small radius.
> Using a carefully designed inductive argument, see Lemma C.5, we establish an attraction rate of $1/\sqrt{T}$, bounded decisions and hence bounded velocities.
>
> **Time-varying constraints**
>
> Here, the previous geometric arguments fail because $x_{T}^*$ does not belong to every feasible set $\mathcal{C}\_{t}$.
> This means that the environment can choose a sequence of costs and constraints such that $-r_{t}^T(x_{T}^*-x_t)>0$ for all $t\in\\{1,\dots,T-1\\}$, making the regret bound vacuous.
> To avoid this issue, we limit the environment as follows.
> We show in Lemma D.3 that the polyhedral intersection assumption suffices to extend the above geometric property to the time-varying constraint setting, i.e., $-r_{t}^T(x_{T}^*-x_t)\leq0$ for all $t$.
> Moreover, the slowly time-varying constraint assumption strikes a balance between the convergence to the feasible set and the power of the environment to select feasible sets that are far away from the initial one (see 4 above).
> In addition, we show in Lemma D.6 that the time-averaged constraints satisfy the slowly time-varying assumption.
>
> 7. "Could you provide some real-world examples where your results can be applied?"
>
> For a detailed answer, we refer the reviewer to the broader implications part of our global response.

---

> > ### Comment · Reviewer_pGFm · 2023-08-14
> >
> > I thank the authors for the detailed response. Thought I still feel the constraints are quite artificial, in light of the reviews from other reviewers and the authors' responses, I'm willing to adjust my rating upwards.

---

### Official Review · Reviewer_TfpV · 2023-07-06

**Soundness:** 4 excellent
**Presentation:** 4 excellent
**Contribution:** 4 excellent
**Rating:** 7
**Confidence:** 4

**Summary:**

The paper studies an online learning problem in which the learner aims at maximizing their reward subject to adversarial time-varying nonlinear constraints. The proposed algorithm guarantees a regret upper bound of order $T^{1/2}$ and converges to the feasible set at a rate of $T^{-1/2}$. Intrestingly, the feasible set may be unknown to the learner and slowly change over time. An interesting application described in the paper is computing equilibria in two-players zero-sum games in which players share a common resource.

**Strengths:**

The set-up is very interesting. I particularly appreciated the connection to computing GNEs of some kind. The techniques introduced are novel and may find applications also in related problems.

**Weaknesses:**

I would add to the related works, when discussing performance metrics, also some pointers to work on online learning with long-term constraints stemming from the literature on bandits with knapsacks. One example is: "A Unifying Framework for Online Optimization with Long-Term Constraints" by Castiglioni et al. (NeurIPS 22) It would be nice to see a discussion of their baselines and how they relate to yours.

**Questions:**

None.

**Limitations:**

N.A.

---

> ### Author Rebuttal · Authors · 2023-08-09
>
> Thank you for your valuable suggestions and constructive feedback.
> We answer the raised questions below.
>
> **Weaknesses**
>
> 1. "I would add to the related works, when discussing performance metrics, also some pointers to work on online learning with long-term constraints stemming from the literature on bandits with knapsacks."
>
> Following the reviewer's comments, in addition to lines $\\{38,128\\}$ where we mentioned the work of Sun et al. [6] on safety-aware algorithms for adversarial contextual bandits, we will expand the related work section to include a discussion of the ``Bandits with Knapsacks'' framework.
> In particular, we will discuss the works by [1-5], see the references below.
>
> 2. "One example is: "A Unifying Framework for Online Optimization with Long-Term Constraints" by Castiglioni et al. (NeurIPS 22) It would be nice to see a discussion of their baselines and how they relate to yours."
>
> Thanks for the excellent point.
> We have updated the manuscript with a discussion of Castiglioni et al. [2022].
> Next, we bring into context the study by Castiglioni et al. [2022] and discuss how our baseline relates to it.
>
> Castiglioni et al. [2022] studied the following symmetric setting with adversarial environment, baseline $x^{\star}$ belonging to $\arg\min_{x\in\mathcal{X}}\sum_{t=1}^{T}f_{t}(x)$ subject to $\sum_{t=1}^{T}g_{t}(x)\geq0$, iterates $\{x_t\}$ satisfying $\sum_{t=1}^{T}g_{t}(x_{t})\geq-o(T)$, and regret $\sum_{t=1}^{T}f_{t}(x_t)-\sum_{t=1}^{T}f_{t}(x^{\star})$.
> Their baseline and regret formulation can therefore be obtained as a special case of our formulation with time-averaged constraints, that is, when our $g_T(x)$ is chosen as $\frac{1}{T}\sum_{t=1}^T g_t(x)$.
> We further note that Castiglioni et al. [2022] introduce the parameter $\rho=\sup_{x\in\mathcal{X}}\min_{t\in[T]}\min_{i\in[m]}g_{t,i}(x)$, which is required to be positive and known to the algorithm for achieving $\mathcal{O}\_{\rho}(\sqrt{T})$ regret.
> We note that $\rho>0$ implies that the intersection of all feasible sets is non-empty, which is a strong assumption (as is knowledge about the parameter $\rho$).
> In our formulation with time-averaged constraints, Assumption 1.2 reduces to the feasible set $\mathcal{C}\_t$ belongs to a polyhedral intersection $\mathcal{Q}\_t$, which does not require a non-empty intersection of all $\mathcal{C}\_t$ (and has a geometrical interpretation; also note that the assumption $\lVert g_{t}(x)-g_{t-1}(x) \rVert_{\infty} = \mathcal{O}(1/t)$ is automatically satisfied in this case).
> Thus, there are situations, where the regret bound from Castiglioni et al. [2022] becomes vacuous (for $\rho=0$), while our method still provably achieves $\mathcal{O}(\sqrt{T})$ regret.
> Additional differences are that Castiglioni et al. [2022] considers primal-dual methods and assumes that all constraints are revealed after every iteration, whereas our method is primal-only and has only partial information about all violated constraints.
> The latter point reduces computation and simplifies projections onto the velocity polytope, but requires a nontrivial inductive argument for establishing $\sqrt{T}$-regret.
>
> In summary, our paper retains its novelty and importance also in the light of Castiglioni et al. [2022].
>
> **References on ``Bandits with Knapsacks''**
>
> [1] Ashwinkumar Badanidiyuru, Robert Kleinberg, and Aleksandrs Slivkins. Bandits with knapsacks.
> Journal of the ACM (JACM), 65(3):1-55, 2018
>
> [2] Shipra Agrawal and Nikhil R Devanur. Bandits with concave rewards and convex knapsacks. In
> Proceedings of the fifteenth ACM conference on Economics and computation, pages 989-1006.
> ACM, 2014.
>
> [3] Shipra Agrawal and Nikhil R Devanur. Bandits with global convex constraints and objective.
> Operations Research, 67(5):1486-1502, 2019.
>
> [4] Nicole Immorlica, Karthik Abinav Sankararaman, Robert Schapire, and Aleksandrs Slivkins.
> Adversarial bandits with knapsacks. In 60th IEEE Annual Symposium on Foundations of Computer
> Science, FOCS 2019, pages 202-219. IEEE Computer Society, 2019.
>
> [5] Xuanyu Cao and KJ Ray Liu. Online convex optimization with time-varying constraints and
> bandit feedback. IEEE Transactions on automatic control, 64(7):2665-2680, 2018.
>
> [6] Wen Sun, Debadeepta Dey, and Ashish Kapoor. Safety-aware algorithms for adversarial contextual
> bandit. Proceedings of Machine Learning Research, 70:3280-3288, 2017.

---

> > ### Comment · Reviewer_TfpV · 2023-08-21
> >
> > Thank you for the detailed response. I confirm my positive score.

---

### Official Review · Reviewer_9WyZ · 2023-07-12

**Soundness:** 4 excellent
**Presentation:** 3 good
**Contribution:** 3 good
**Rating:** 6
**Confidence:** 2

**Summary:**

This paper considers an online constrained learning scenario, where partial information about the objective and the constraints, which could potentially be time-varying, is available at each time slot. The proposed algorithm, referred to as "Constraint Violation Velocity Projection," constructs a velocity polytope at each time step and updates the decisions based on the solution to a quadratic velocity projection problem. It is shown that the proposed method leads to a $\mathcal{O}(\sqrt{T})$ regret and convergence rate of $\mathcal{O}(1/\sqrt{T})$ to the feasible set. Numerical results are also provided to corroborate the theoretical contributions.

**Strengths:**

- The considered problem is interesting and challenging from the perspective of the limited information available to the learning algorithm and the variation of constraints over time.
- The paper is well written, and the theoretical analyses of the proposed method could be of interest to the NeurIPS audience.

**Weaknesses:**

This is mostly a theoretical piece of work, but it is slightly unclear to me what the broader implications of the proposed algorithm would be. The two-player game experimental results presented in Section 4 are interesting, but I wonder if the authors could highlight at least another example use-case of the proposed algorithm.

**Questions:**

- How does the proposed method compare with primal-dual Lagrangian approaches? Do these approaches completely fail if the feedback received from the environment is limited, as assumed in this work?
- Are there applications of the proposed algorithm in the context of constrained reinforcement learning with time-varying constraints and (potentially) partial observability? Following my comment under "**Weaknesses**," this might be an area where the proposed algorithm could shine.

**Limitations:**

Please refer to my comments under "**Weaknesses**" regarding the broader impact and application of the proposed algorithm.

---

> ### Author Rebuttal · Authors · 2023-08-09
>
> Thank you for your valuable suggestions and important remarks.
> We answer the raised questions below.
>
> **Weaknesses**
>
> "This is mostly a theoretical piece of work, but it is slightly unclear to me what the broader implications of the proposed algorithm would be.
> The two-player game experimental results presented in Section 4 are interesting, but I wonder if the authors could highlight at least another example use-case of the proposed algorithm."
>
> As pointed out by the reviewer, our work is theoretical and its the main contribution is to design and analyze a novel algorithm that combines techniques from the seemingly distant fields of online convex optimization (online gradient descent) and non-smooth mechanics (velocity space).
> However, we agree that potential applications are important.
> Here we provide a few and give a more detailed explanation in answering to question 2 below:
> adversarial contextual bandits with sequential risk constraints [Sun et al., 2017], network resource allocation [Chen et al., 2017], logistic regression [Cao and Liu, 2019, Liu et al., 2022], ridge regression and job scheduling [Liu et al., 2022], 2-player games with resource constraints (Section 4), system identification and optimal control.
>
> We will update the manuscript accordingly.
>
> **Questions**
>
> 1.1 "How does the proposed method compare with primal-dual Lagrangian approaches?"
>
> There are two major differences between previous work on unknown and slowly time-varying constraints and ours.
> The problem formulation in Chen et al. [2017], Neely and Yu [2017], Yu et al. [2017], Sun et al. [2017], Liu et al. [2022] differs in the definition of performance metrics in terms of the benchmark(s) used in the regret and constraint violation criteria, see our response to 7SGB Weaknesses 1.
> Moreover, it assumes that on each time step all constraints are fully revealed to the learner.
> More specifically, the above primal-dual approaches maintain a dual variable $\lambda_i\geq0$ throughout the execution of the algorithm, for every index $i\in\\{1,\dots,m\\}$ corresponding to constraints $\\{g_{t,i}\\}\_{t\geq1}$.
> That is, in time step $t$, updating the dual variables $\lambda$ requires information for all constraints $\\{g_{t,i}\\}\_{i=1}^{m}$.
>
> In contrast to prior work, we consider the setting in which *only partial information about the all violated constraints* is revealed to the learner.
> Furthermore, the velocity polyhedral projection allows us to design a primal-only algorithm, that enjoys the following properties:
> i) it requires partial information only for all violated constraints (reducing the number of constraints); and
> ii) the velocity projection can be computed efficiently since the constraints are always linear.
>
> We will make sure that this difference is better discussed in the paper.
>
> 1.2 "Do these approaches completely fail if the feedback received from the environment is limited, as assumed in this work?"
>
> The short answer is: the primal-dual Lagrangian approach does not directly extend to the setting of constraint violation oracle, since updating the dual variables requires information for all constraints, in every iteration.
> However, it is an interesting open question to extend the primal-dual Lagrangian approach to the setting of only partial information for all violated constraints.
>
> 2. "Are there applications of the proposed algorithm in the context of constrained reinforcement learning with time-varying constraints and (potentially) partial observability? Following my comment under "Weaknesses," this might be an area where the proposed algorithm could shine."
>
> This is an excellent point, which we will discuss in the paper.
> Here are more details about a potential application to system identification and optimal control, where an agent must predict a sequence of actions to minimize costs and satisfy constraints.
> Many real-world systems are subject to wear, tear and drift (e.g., sensors), which naturally leads to non-stationary costs and constraints, corresponding to slowly time-varying functions $f_t$ and $\\{g_{t,i}\\}\_{i=1}^{m}$, respectively.
> Furthermore, it is common in optimal control to know analytically both the dynamics model and the cost and constraint functions, so the gradients are naturally available.
> Assuming access to a constraint violation oracle, the above scenario can be cast into our online problem formulation.
> More specifically, in each episode $t$, an agent $\phi$ parameterized by weights $\theta_t\in\mathbb{R}^n$ generates a sequence of actions $\\{x_{\ell}\\}\_{\ell=1}^{H}$ and upon their deployment in the environment, receives a cost value $f_t(\theta_t)$, gradient $\nabla f_t(\theta_t)$ and information for all violated constraints $\\{(g_{t,i}(\theta_t), \nabla g_{t,i}(\theta_t)\\}\_{i\in I(\theta_t)}$.

---

> > ### Comment · Reviewer_9WyZ · 2023-08-14
> > **Thank you!**
> >
> > Thank you very much for your response to my comments. In light of the authors' rebuttal and other reviewers' feedback, I have decided to maintain my score, favoring acceptance.

---

### Official Review · Reviewer_7SGB · 2023-07-12

**Soundness:** 3 good
**Presentation:** 2 fair
**Contribution:** 3 good
**Rating:** 5
**Confidence:** 2

**Summary:**

The paper studies a constrained online learning problem. The considered problem has a notion of constraint violation which is different from previous online learning problems. Also, the environment chooses not only a cost function but also a constraint function at each round. The algorithm can observe the cost function and the violated constraint function after playing each action. The authors propose an algorithm called CVV-Pro to solve this problem for both time-invariant and adversarial nonlinear constraints. The algorithm projects the gradients into a constructed velocity polytope in each step. The analysis shows that regret is sublinear and the constraint violation decreases with $O(1/\sqrt{t})$ for each round $t$.

**Strengths:**

 The paper considers a novel constrained online learning problem where the environment chooses both the cost function and the constraint function. The proposed algorithm does not require the knowledge of the horizon length $T$ in advance. The paper bounds a new notion of the the constraint violation. The conclusions and the proof sketches are clearly presented. The proposed algorithm may be of interest by the community.

**Weaknesses:**

The paper can be improved in the following points:

- The time-varying constraints are also considered in [1,2,3]. Thus, it is not appropriate to claim that the setup differs from traditional OCO in terms of the time-varying constraint function.

- Although the considered constraint in (1) requires the constraint for the last round. The previous cumulative constraints [1] should also imply this.

- The bound of constraint violation relies on Assumption 1.2 which requires the change of constraint function shrinks with time and the constraint set $\mathcal{C}_t$ is contained in $\mathcal{Q}_t$. It would be better to explain why the two assumptions are reasonable.



[1] Neely, M.J. and Yu, H., 2017. Online convex optimization with time-varying constraints. arXiv preprint arXiv:1702.04783.
[2] Yu, H., Neely, M. and Wei, X., 2017. Online convex optimization with stochastic constraints. Advances in Neural Information Processing Systems, 30.
[3] Cao, X. and Liu, K.R., 2018. Online convex optimization with time-varying constraints and bandit feedback. IEEE Transactions on automatic control, 64(7), pp.2665-2680.

**Questions:**

Please see the questions in the Weakness part. Other questions are listed as below.

1. Does the constraint violation bound cover the cumulative constraint violation bound in [1]?

2. In Assumption 1.2, why can $\mathcal{C}_t$ be contained in $\mathcal{Q}_t$?

3. In Theorem 3, what is the intuition of the attraction?

**Limitations:**

The authors may need to address why  Assumption 1.2 and Assumption 3.1 are reasonable.

---

> ### Author Rebuttal · Authors · 2023-08-09
>
> Thank you for your valuable suggestions and important remarks.
> We answer the raised questions below.
>
> **Weaknesses**
>
> 1. "The time-varying constraints are also considered in [1,2,3]. Thus, it is not appropriate to claim that the setup differs from traditional OCO in terms of the time-varying constraint function."
> While exhibiting certain similarities upon initial observation, prior performance measures are quantitatively different.
> On the one hand, the work by  Chen et al. [2017], Cao and Liu [2019], Liu et al. [2022] use $\sum_{t=1}^{T} f_t(x_t) - \sum_{t=1}^{T} f_t(x_{t}^{\star})$ as a performance measure, where the iterates $x_t$ are required to satisfy $\sum_{t=1}^{T} g_t(x_t) \geq -c \sqrt{T}$ and the optimal solutions $\\{x\_{t}^{\star}\\}\_t$ satisfy $x_{t}^{\star}\in\arg\min_{x}\\{f_t(x) : g_t(x)\geq 0\\}$.
> On the other hand (lines 123-139), the work by Neely and Yu [2017], Yu et al. [2017], Sun et al. [2017]  use $\sum_{t=1}^{T} f_t(x_t)- \sum_{t=1}^{T} f_t(x^{\star})$ as a performance measure, where the iterates $x_t$ are required to satisfy $\sum_{t=1}^{T} g_t(x_t) \geq -c \sqrt{T}$ and the optimal solution $x^{\star}$ satisfies $g_t(x^{\star})\geq 0$ *for all* $t\in\\{1,\dots,T\\}$.
> This leads to a major asymmetry in the way regret is measured: while the iterates of the online algorithm only need to satisfy a cumulative measure of constraint violation, the benchmark $x^{\star}$, which represents the best fixed decision in hindsight, is required to satisfy *all* constraints $g_t(x^{\star})\geq 0$ for $t=\\{1,\dots,T\\}$.
> In contrast, our performance metric is symmetric and imposes the same constraints on the learner as well as the benchmark $x^*$ (soundness is ensured by the slowly time-varying assumption).
> In particular, for every $T\geq1$ (as is unknown), let $\mathcal{C}\_T:=\\{x\in \mathbb{R}^n\~|\~g_T(x)\geq0\\}$, then we seeks to optimize $\sum\_{t=1}^{T} f\_t(x\_t) - \min\_{x^* \in \mathcal{C}\_T} \sum\_{t=1}^{T} f\_t(x^*)$ subject to $g_T(x_T)\geq - \frac{c}{\sqrt{T}}$.
>
> 2. "Although the considered constraint in (1) requires the constraint for the last round.
> The previous cumulative constraints [1] should also imply this."
> We respectfully disagree.
> As explained above Neely and Yu [2017] focuses on time-varying constraints, where, however, the following weaker notion of constraint violation is used: $\sum_{t=1}^{T} g_t(x_t) \geq -c \sqrt{T}$.
> This metric allows constraint violations for many iterations, as long as these are compensated by strictly feasible constraints (*in the worst case even with a single feasible constraint with a large margin*).
>
> 3. "The bound of constraint violation relies on Assumption 1.2 which requires the change of constraint function shrinks with time and the constraint set $\mathcal{C}_t$ is contained in $\mathcal{Q}_t$. It would be better to explain why the two assumptions are reasonable."
> See the Assumption part of the global response.
>
> **Questions**
>
> 1. "Does the constraint violation bound cover the cumulative constraint violation bound in [1]?"
> Our constraint violation bound $g_T(x_T)\geq - \frac{c}{\sqrt{T}}$ for all $T\geq1$, implies the cumulative constraint violation bound in Neely and Yu [2017]
> $\sum_{t=1}^{T}g_{t}(x_{t})\geq-\sum_{t=1}^{T}\frac{c}{\sqrt{t}}\geq-2c\sqrt{T}$.
>
> 2. "In Assumption 1.2, why can $\mathcal{C}_t$ be contained in $\mathcal{Q}_t$?"
> See the Assumption part of the global response.
>
> 3. "In Theorem 3, what is the intuition of the attraction?"
> As explained in lines 271-279, we add an additional hypersphere constraint $g_{m+1}(x_{t})=\frac{1}{2}[R^{2}-\Vert x_{t}\Vert^{2}]$ that attracts any decision sequence $\\{x_{t}\\}\_{t\geq1}$ generated by the learner to a hypersphere $\mathcal{B}\_{R}$ and guarantees that it always stays inside a hypersphere $\mathcal{B}_{4R}$ with a slightly larger radius.
> In fact, see lines 28-39 in the supplementary, we provide a refined analysis, reducing the radius $70R$ to $4R$, which uses step sizes of the form $\eta_t=1/(\alpha\sqrt{t + d})$, for any constant $d\geq0$.

---

> > ### Comment · Reviewer_7SGB · 2023-08-14
> > **Thank you for the rebuttals**
> >
> > I thank the authors for the response to my questions.
> >
> > I am convinced that the constraint considered in this work is more strict than the cumulative ones, so I am willing to increase the score. I would recommend the authors to correct the typo in Assumption 1.2 and add the discussions about the assumptions in the paper.

---

### Official Review · Reviewer_dZ67 · 2023-07-25

**Soundness:** 3 good
**Presentation:** 3 good
**Contribution:** 3 good
**Rating:** 5
**Confidence:** 2

**Summary:**

CVV-Pro is an online learning framework for non-stationary data streams, with support for handling constraints that may be unknown,  adversarially time-varying, and nonlinear. It doesn't assume that the feasible set is known, but proves that the algorithm still converges to it a rate of 1/sqrt(T). The method is designed to target complex online optimization and adaptive decision making tasks, with applications in two player games with shared constraints.

The method makes a few assumptions:
1. It allows constraints to be relaxed and violated. Instead of ensuring feasibility at each step, it makes the violations converge to zero at a rate of O(1/sqrt(T)).
2. An oracle exists that returns the value and the gradient for all the violated constraints.
3. Constraints are allowed to vary in time, but at a decreasing rate of change at <= O(1/t)

Under these constraints, the paper proves that optimal regret at O(sqrt(T)) can be achieved. It is shown to be more efficient than online methods that project the candidates onto the constraint manifold (feasible set), but this may be expected since this method allows for the constraints to be actually violated, which would certainly reduce the loss faster.


**Strengths:**


This is my understanding of the method: Similar to online gradient descent, the method gets the losses associated with x_t. But instead of projecting on the right constraint manifold (because we may not know it), the paper assumes that we can compute or get the gradients for each constraint. While the gradients could also be used directly (by turning constraints into soft regularizers), instead the method constructs a polytope from points that are shifted versions of x_t, by the value of the feasibility loss (g_t) in the direction of its gradient, scaled by a hyperparameter. So, in essence it is very similar to gradient descent (which would simply take the average of these points), but then it effectively projects the loss gradient onto this polytope to retrieve an update direction, which is then scaled by a step size and added to x_t.

This may go around the issue of having very complex constraint manifolds, where a direct projection may not be possible.

**Weaknesses:**

1) The paper is not easy to read, as a lot of the variables and concepts are not introduced linearly.
2) Assumptions may be making the applicable problems very artificial, which takes away from the impact
3) Figures need a bit more polish as they help a lot with understanding the main idea behind the paper.
4) Experiments are pretty weak.
5) No comparison with pure gradient descent

Details:
1) The paper is not easy to read, as it sometimes skips introducing crucial variables. The reader needs to read ahead and jump back and forth to understand everything. For instance, g_t(x) is not explicitly defined on L30 (although clear from context later on). The reader can only guess that it is some sort of feasibility function. Confusingly x^* doesn't have a time index in Eq 1. L_F is not introduced in L50, presumably a smoothness index. B_70R in L50 is not explained (where does 70 come from?). The paper mentions that these assumptions are standard in online optimization literature, but skipping these details makes it hard for a new reader to understand the paper.

2) It's very hard to make sense of all the assumptions the paper makes. Does allowing to violate constraints make sense? Does it make sense to assume that we have access to the gradients of the constraints? Does allowing the constraints to vary at decreasing rates correspond to any realistic setting for a practical problem? It feels like the paper is attempting to come up with an optimal algorithm for a problem that doesn't exist.

3) I didn't understand Figure 1 at all. Figure 2 does almost a good job in explaining the method, but it is too imprecise. The arrows look hand drawn, and the actual area that we should focus on is very small.

4) Experiments consist of artificially created two player "games", simulated a handful of times, and compared against a method that does assume constraints should not be violated. It's not clear to me where the real impact lies. The paper mentions tons of related work, including ones it was "inspired from", but there is no apple to apple comparison of any sort. The only results show that for the artificial setting the authors picked, their proof is consistent with the simulated results.

5) Since the method has access to the gradients, and allows violating the "constraints", a much simpler baseline would be to simply add the constraint gradient losses to the gradient descent. This is already commonly done in other fields, and allows for efficient optimizers like Adam to be used. Since they also have gradients, constraints can also be time varying and nonlinear, with unknown feasible sets. Without comparing against this simple and powerful baseline, I'm not sure I understand the value of this method.

For instance, spectral normalization of the operators is known to help with training GANs, and it can either be done by explicit normalization, or by adding soft regularizers that pull the solution toward the right manifold. The papers that try these two methods are:
Regularizer: (Yoshida & Miyato 2017) " Spectral norm regularization for improving the generalizability of deep learning"
Normalizer: (Miyato et. al) "Spectral normalization for Generative Adversarial Networks".
The second paper mentions how normalization (projection) results in sample data dependent gradients unlike the regularizer (Section 2.1), which proves to be superior. This would be an interesting framework to compare the proposed method to these baselines.

**Questions:**

1) I don't understand 2.ii in L61. If Q_t = U(Q_t, S_t), and Q_0 is already R^n, how does this tell us anything useful? It just keeps getting a union of R^n with its subsets, which makes all Q_t's equal to R^n? I don't understand how Fig 1 clarifies this. I think either the union is wrong, or the initial condition. 2.i in L60: were these supposed to be g_t(x_t) instead of g_t? This makes it look like g_t's are converging, but the paper says in L68 that we should "Note that ||g_t-g_1|| = O(ln(t))", so it's not necessarily converging (but how do we "note" that?). The paper says that assumption 2.ii avoids this situation, but because I don't understand what 2.ii tries to achieve with the union, I don't get this part. I understand in essence what the paper tries to do, but I can't make what I read match what I understand.

2) What is the difference from using the constraint gradients directly to update x_t a la gradient descent? Why do you think this is a better method?

3) Why not make online gradient descent also relaxed so that projections are allowed to violate the constraints by a threshold, provided losses are better? That would give a more apple to apple comparison.


**Limitations:**

Like I mentioned above, I think the main limitation comes from all the assumptions (constraint have accessible gradients, vary at a decreasing rate, and can be violated). As the paper only provides a couple of artificial two player games in experiments, it's not clear how limited the actual impact of this work is.

---

> ### Author Rebuttal · Authors · 2023-08-09
>
> Thank you for your valuable suggestions and constructive feedback.
> We answer the raised questions below.
>
> **Weaknesses**
>
> 1.1 "The paper is not easy to read ... can only guess that it is some sort of feasibility function."
> Following the reviewer's comments, we have reorganize and improved the presentation in lines 29-31: "... constraints, governed by the following interaction protocol (see Assumption 1.2).
> In each time step $t$, ... $\mathcal{C}\_t:=\\{x\in \mathbb{R}^n\~|\~g\_t(x)\geq0\\}$, where the constraint function $g_t(x)$ is defined as $[g_{t,1}(x),\dots,g_{t,m}(x)]$."
>
> 1.2 "Confusingly $x^*$ doesn't have a time index in Eq 1."
> Although it is a standard notation in online convex optimization [Hazan 2016] to omit the time dependence for the optimal solution, to improve the clarity, we will add the time index notation.
>
> 1.3 "$L_F$ is not introduced in L50, presumably a smoothness index."
> Assumption 1.1 now reads "There exist $R,L_{\mathcal{F}},L_{\mathcal{G}}>0$ ...
>
> 1.4 "$B_{70R}$ in L50 is not explained (where does 70 come from?)."
> In Section 3.2, we algorithmically enforce that any decision sequence generated by the learner is attracted to a hypersphere of radius $R$ and more importantly always stays inside a hypersphere with a slightly larger radius $4R$, see lines 278-279.
> (In fact, see lines 28-39 in the appendix, we provide a refined analysis that reduces the radius $70R$ to $4R$.)
> This property is crucial for establishing a uniform bound on the velocity, which in turn allows us to obtain a $\sqrt{T}$ regret (lines 238-239).
> We added a short discussion with the appropriate reference.
>
> 2.1 "Does allowing to violate constraints make sense?"
> The constraints are unknown, time-varying and only partial information about them is revealed to the learner.
> Moreover, the learner cannot ensure that $x_t\in\mathcal{C}_t$, since the environment reveals a feasible set $\mathcal{C}_t$ once $x_t$ is committed.
>
> 2.2 "Does it make sense to assume that we have access to the gradients of the constraints?"
> Unlike previous work (lines $\\{119,120,123\\}$), which requires full knowledge of all constraint functions, CVV-Pro requires only partial information for all *violated* constraints.
> In addition, it seems natural to us that the environment can compute derivatives of the violated constraints, as they are typically known analytically.
>
> 2.3 "Does allowing the constraints to vary at decreasing rates correspond to any realistic setting for a practical problem?"
> All previous work (line 123) makes certain slowly time-varying assumptions.
> For further discussion, see our response 2. to (TfpV) and 4. to (pGFm).
>
> 2.4 "It feels like the paper is attempting to come up with an optimal algorithm for a problem
> that doesn't exist."
> We give the first online algorithm for the challenging setting where the constraints are unknown, slowly time-varying, and only partial information about their violations is revealed.
> There are already interesting applications, as mentioned in our global response.
> In addition, we believe that new applications will emerge, similar to the applications of bandits that were first studied in theory decades ago and later applied in practice in areas such as recommendation systems and Internet traffic routing.
>
> 3 "I didn't understand Fig.1 at all. Fig.2 does almost a good job..."
> An additional figure for explaining the Assumption is presented in the rebuttal pdf file (which will likely only fit into the appendix) and a modified version of Fig.2 is also provided.
>
> 4 "artificially created two player games; where the real impact lies; no apple to apple comparison..."
> It is important to note that the constraints are unknown and only partial information (violations) about them is revealed to the learner.
> In contrast to primal-dual methods (in related work), we give a primal-only algorithm that tackles a different problem formulation using only partial information.
> We cannot compare to them, as these methods are not applicable.
> See also answer to Question 1 of (9WyZ).
>
> 5 "a much simpler baseline would be to simply add the constraint gradient losses to the gradient descent..."
> The short answer is: standard online optimization methods cannot efficiently tackle this setting. In particular, in penalty methods the multiplier for constraint violations might need to be chosen arbitrarily high such that regret bound will be vacuous.
> However, a recent line of work tackles this problem algorithmically, but their scope is different: either they consider time-invariant constraints (lines 119-120) or a non-symmetric notion of constraint satisfaction and benchmark (lines 123-139).
> See also answer to Question 1.1 to (9WyZ).
>
> **Questions**
>
> 1.1 "I don't understand 2.ii in L61 ... either the union is wrong, or the initial condition."
> "... I don't understand what 2.ii tries to achieve with the union ..."
> We apologize for the typo. The union should be replace with intersection, see the Assumption part of the global response.
>
> 1.2 "2.i in L60: were these supposed to be $g_t(x_t)$ instead of $g_t$?"
> It now reads $\Vert g_{t}(x)-g_{t-1}(x)\Vert_{\infty} =\mathcal{O}(1/t)$ for all $x\in\mathcal{B}_{4R}$.
>
> 1.3 "This makes it look like $g_t$'s are converging, but ..."
> The series $\sum_{i=1}^{n}1/i \approx \ln(n)$ diverges.
> Further justification are provided in our response 4. to (pGFm).
>
> 2. "What is the difference from using the constraint gradients directly to update $x_t$ a la gradient descent? Why do you think this is a better method?"
> i) See our response to Weaknesses 5.
> ii) CVV-Pro is a  primal-only algorithm with non-asymptotic regret and convergence guarantees, i.e., for every iteration $t$.
> Also, the velocity polytope (due to linear constraints) admits an efficient projection computation.
>
> 3. "Why not make online gradient descent also relaxed ... That would give a more apple to apple comparison."
> See our response to Weaknesses 5.

---

> > ### Comment · Reviewer_dZ67 · 2023-08-10
> >
> > Thank you very much for the careful and detailed responses, and all the changes made to the paper. Also both figures are vastly improved I think. Having read all the responses and discussions, I have less concerns about the applicability of the method regarding the assumptions it makes, and will change my score accordingly.

---

### Author Rebuttal · Authors · 2023-08-09

We would like to thank the reviewers for the time they spent reviewing our paper and for their insightful comments.

Several reviewers recognized the novelty of the setting studied and the theoretical guarantees provided, as well as the applicability of our algorithm to related problems.
More specifically:
[problem formulation] "novel constrained online learning problem" (7SGB), "set-up is very interesting" (TfpV), "interesting and challenging from the perspective of the limited information" (9WyZ);
[technical results] "prove some interesting technical results" (pGFm), "bounds a new notion of the constraint violation" (7SGB), "the conclusions and proof sketches are clearly presented" (7SGB);
[algorithm] "effectively projects the loss gradient onto the velocity polytope" (dZ67), "does not require the knowledge of the horizon length in advance" (7SGB).
[application] "appreciated the connection to computing GNEs" (TfpV), "techniques introduced are novel and may find applications also in related problems" (TfpV), "proposed algorithm may be of interest by the community" (7SGB), "theoretical analyses of the proposed method could be of interest to the NeurIPS audience" (9WyZ).

Thank you for the positive feedback.

The main concerns raised relate to the assumptions (dZ67,7SGB,pGFm) and the broader implications of the proposed method (9WyZ,TfpV):

**Assumptions**

We regret that there was a typo in Assumption 1.2 Part 2, the union symbol should have been an intersection, this caused unnecessary confusion. The corrected version reads:

**(Updated)** the environment chooses $f_t\in \mathcal{F}$ and $g_t\in\mathcal{G}$ such that i) $\Vert g_{t}(x)-g_{t-1}(x)\Vert\_{\infty} =\mathcal{O}(1/t)$ for all $x\in\mathcal{B}\_{4R}$, with $\Vert\cdot\Vert\_{\infty}$ the $\ell\_{\infty}$ norm, and ii) $\mathcal{C}\_t$ is contained in $\mathcal{Q}\_t:=\cap_{\ell=0}^{t-1}\mathcal{S}\_{\ell}$, where $\mathcal{S}\_t:=\\{x\in \mathbb{R}^n \~|\~G(x_t)^{\top}(x-x_t)\geq0 \\}$  is a cone centered at $x_t$ for $t\geq1$ and $\mathcal{S}_{0}=\mathbb{R}^{n}$.

(The situation is illustrated in the rebuttal pdf file and will be added to supplementary.)

The reviewers had concerns about the novelty and necessity of the assumptions.
It is important to note that we are considering an online learning problem where the agent is dealing with unknown and time-varying constraints, and the environment reveals information for them after the learner has made a decision.
In contrast to previous work, where all constraints are fully revealed to the learner at each time step, we focus on the more practical setting where the learner has access to only *partial* information about all *violated* constraints.
For this challenging setting, we give the first online algorithm that enjoys provable guarantees and efficient projection computation due to the linear structure of the velocity polytope.

It is crucial to emphasize that in the above setting one has to make assumptions to obtain non-trivial regret and convergence guarantees.
Further justification is provided in our response 2. to (TfpV) and 4. to (pGFm).

We have updated the manuscript to reflect this discussion, as elaborated in the individual responses.

**Broader implications**

It is important to note that our work is theoretical, and the main contribution is to design and analyze a novel algorithm that combines techniques from the seemingly distant fields of online convex optimization (online gradient descent) and non-smooth mechanics (velocity space).
Nevertheless, the list of potential applications includes, but is not limited to: adversarial contextual bandits with sequential risk constraints [Sun et al., 2017], network resource allocation [Chen et al., 2017], logistic regression [Cao and Liu, 2019, Liu et al., 2022], ridge regression and job scheduling [Liu et al., 2022], 2-player games with resource constraints (Section 4), system identification and optimal control.

More specifically, let us elaborate on system identification and optimal control application, where an agent must predict a sequence of actions to minimize costs and satisfy constraints.
Many real-world systems are subject to wear, tear and drift (e.g., sensors), which naturally leads to non-stationary costs and constraints, corresponding to slowly time-varying functions $f_t$ and $\\{g_{t,i}\\}\_{i=1}^{m}$, respectively.
Furthermore, it is common in optimal control to know analytically both the dynamics model and the cost and constraint functions, so the gradients are naturally available.
Assuming access to a constraint violation oracle, the above scenario can be cast into our online problem formulation.
More specifically, in each episode $t$, an agent $\phi$ parameterized by weights $\theta_t\in\mathbb{R}^n$ generates a sequence of actions $\\{x_{\ell}\\}\_{\ell=1}^{H}$
 and upon their deployment in the environment, receives a cost value $f_t(\theta_t)$, gradient $\nabla f_t(\theta_t)$ and information for all violated constraints $\\{(g_{t,i}(\theta_t), \nabla g_{t,i}(\theta_t)\\}_{i\in I(\theta_t)}$.

We will incorporate this discussion about the applicability.

We believe our answers and changes detailed in the individual responses clarify the questions and address the raised concerns about our work.

---

### Decision · Program_Chairs · 2023-09-21

**Decision:**

Accept (poster)

**Comment:**

This work proposes a method for online learning under adversarial, time-varying, and nonlinear constraints. Under the assumption that the constraints are slowly changing, and given an oracle that reveals partial information of the violated constraints, the CVV-Pro method obtains $O(\sqrt{T})$ regret and converges to the latest feasible set at a rate of $O(1/\sqrt{T})$. The method is then applied to obtain equilibria for two-player zero-sum games. During the discussion, concerns were raised regarding presentation, the assumptions on the slowly-changing constraints, and the broader applicability of the algorithm. These concerns were adequately addressed for the most part, and given the new formulation of the problem and the novel algorithm, I recommend acceptance.